# Synergistic Interaction between Symbiotic N_2_ Fixing Bacteria and *Bacillus strains* to Improve Growth, Physiological Parameters, Antioxidant Enzymes and Ni Accumulation in Faba Bean Plants (*Vicia faba*) under Nickel Stress

**DOI:** 10.3390/plants11141812

**Published:** 2022-07-09

**Authors:** Mohssen Elbagory, Sahar El-Nahrawy, Alaa El-Dein Omara

**Affiliations:** 1Department of Biology, Faculty of Science and Arts, King Khalid University, Mohail 61321, Assir, Saudi Arabia; mhmohammad@kku.edu.sa; 2Agricultural Research Center, Department of Microbiology, Soils, Water and Environment Research Institute, Giza 12112, Egypt; sahar.elnahrawy@yahoo.com

**Keywords:** nickel, plant growth-promoting rhizobacteria, physiological modifications, antioxidant enzymes, faba bean, plant stresses

## Abstract

Several activities in the agriculture sector lead to the accumulation of Nickel (Ni) in soil. Therefore, effective and economical ways to reduce soil bioavailability of Ni must be identified. Five isolates of *Rhizobium leguminosarum biovar Viceae* (ICARDA 441, ICARDA 36, ICARDA 39, TAL–1148, and ARC–207) and three bacterial strains (*Bacillus subtilis*, *B. circulance*, and *B. coagulans*) were evaluated for tolerance and biosorption of different levels of Ni (0, 20, 40, 60, and 80 mg L^−1^). Pot experiments were conducted during the 2019/2020 and 2020/2021 seasons using four inoculation treatments (inoculation with the most tolerant *Rhizobium* (TAL–1148), inoculation with the most tolerant *Rhizobium* (TAL–1148) + *B. subtilis*, inoculation with the most tolerant *Rhizobium* (TAL–1148) + *B.* *circulance*, and inoculation with the most tolerant *Rhizobium* (TAL–1148) + *B.* *coagulans*) under different levels of Ni (0, 200, 400, and 600 mg kg^−1^), and their effects on growth, physiological characteristics, antioxidant enzymes, and Ni accumulation in faba bean plants (*Vicia faba* C.V. Nobaria 1) were determined. The results showed that *Rhizobium* (TAL–1148) and *B. subtilis* were the most tolerant of Ni. In pot trials, inoculation with the most tolerant *Rhizobium* TAL–1148 + *B. subtilis* treatment was shown to be more effective in terms of growth parameters (dry weight of plant, plant height, number of nodules, and N_2_ content), and this was reflected in physiological characteristics and antioxidant enzymes under 600 mg kg^−1^ Ni compared to the other treatments in the 2019/2020 season. In the second season, 2020/2021, a similar pattern was observed. Additionally, lower concentrations of Ni were found in faba bean plants (roots and shoots). Therefore, a combination of the most tolerant *Rhizobium* (TAL–1148) + *B. subtilis* treatment might be used to reduce Ni toxicity.

## 1. Introduction

Faba bean (*Vicia faba* L.) is one of the most important leguminous crops grown in Asia and the Mediterranean region [1]. It is high in protein (25–30%) and carbohydrates (55–60%), which contributes to its placement among the popular annually produced grain crops for use among humans and domestic animals [2]. Green seeds are utilized in fresh vegetable salads during vegetative growth, while dry seeds are used in prepared food, and the entire plant can be fed to farm animals [3]. Due to a lack of domestic production, Egypt is one of the leading importers of faba bean [1].

Some stresses, such as heavy metal contamination, have an impact on faba bean productivity. In addition, human, agricultural, and industrial activities all contribute to metal contamination of soils [4]. As a result of these processes, mineral residues accumulate in agricultural soils, posing a threat to food safety and public health [5]. Since microbial flora composition and microbial activity are greatly affected by mineral accumulation, soil fertility is lost [6]. Some metals, while necessary in small amounts for organisms, are toxic in large amounts. One of the most significant environmental and biological issues is nickel (Ni) contamination [7], which is one of the most common trace metals discharged into the environment by both natural and manmade activities. Anthropogenic activities, such as burning fossil fuels for electricity production, mining, smelting, automobile emissions, steel manufacturing, the cement sector, and domestic, municipal, and industrial waste disposal, all contribute to increased Ni release into the soil [8]. In the metallurgical and electroplating sectors, Ni is used as a raw material. It is also employed as a catalyst in the chemical and culinary industries, in addition to being used as a battery backup [8,9]. The release of Ni into the environment, including its deposition in agricultural soils, is a major problem [8,10]. Nickel is a common heavy metal found in soil and water, accounting for around 0.08 percent of the earth’s crust [11]. Nickel toxicity poses a serious threat to agriculture, the environment, and human health [12].

Excess Ni in plants has become a major issue, posing a serious threat to the sustainability of agriculture. Species and age of plant, growing conditions, Ni concentration, and exposure period in the soil all influence the impact of Ni toxicity on physiological and metabolic functions [13,14].

Nickel is required for the synthesis of hydrogenase in prokaryotes, which catalyzes the oxidation of hydrogen liberated by nitrogenase during the dinitrogen reduction process [15]. Nickelin (HypB), an accessory protein responsible for Ni supply in rhizobia, has a dual role in Ni mobilization into hydrogenase and Ni storage [16]. Metals have been shown to negatively affect microorganism growth, morphology, and activity, including symbiotic nitrogen fixation [17]. This symbiosis has been suggested as a method to remove or fix heavy metals in polluted soil and increase the fertility of soil [5]. As a result, finding plant growth-promoting rhizobacteria (PGPR) with high heavy metal resistance capacities became a top priority [18]. Edulamudi [14] showed that, in soils amended with Ni, horse gram coupled with rhizobia could develop nodules and fix nitrogen, and both root nodules and soil were used to assess the rhizobial strains’ biosorption capability for the removal of Ni from contaminated soils. On the other hand, *Bacillus thuringiensis* 002, *B. subtilis* 174, and *B. fortis* 162 accelerated root elongation and Ni mobility in soil and increased Ni accumulation in *Acrasis rosea* [13]. The goal of this study was to examine the Ni stress tolerance and biosorption capability of rhizobia and *Bacillus* strains in their association with faba bean plants under greenhouse conditions during 2019/2020 and 2020/2021 seasons.

## 2. Results

### 2.1. Assessment of Different Rhizobium Isolates and Bacillus Strains for Ni Tolerance

When cultivated in a Yeast Extract Mannitol Broth medium (YEMB) for *Rhizobium* isolates and Nutrient Broth (NB) medium supplemented with varying doses of Ni (0, 10, 20, 40, 60, and 80 mg L^−1^), the growth patterns of *Rhizobium leguminosarum* bv. *Viceae* isolates (ICARDA 441, ICARDA 36, ICARDA 39, TAL–1148, and ARC–207) and *Bacillus* strains (*B. subtilis*, *B. circulance*, and *B. coagulans*) showed substantial change after 72 h. The optical density (OD540) of different bacteria decreased with increasing Ni concentration when compared to normal growth (no Ni). Compared to the other bacteria under study, the TAL–1148 isolate and *B. subtilis* were the most tolerant of higher applied Ni concentrations, showing good abilities to grow on YEMB and NB medium supplemented with 80 mg L^−1^ Ni, achieving 2.53 and 0.84 log numbers, respectively (Table 1).

### 2.2. Biosorption of Ni by Different Rhizobium Isolates and Bacillus Strains

To gain insight into the biosorption of different amounts of Ni by the studied *Rhizobium leguminosarum* bv. *Viceae* isolates (ICARDA 441, ICARDA 36, ICARDA 39, TAL–1148, and ARC–207) and *Bacillus* strains (*B. subtilis*, *B. circulance*, and *B. coagulans*), we used an atomic absorption spectrophotometer to quantify it in a supernatant (Figure 1). The biosorption of the TAL–1148 isolate was the highest among the five isolates evaluated, with 38.83 mg L^−1^ at 80 mg L^−1^. Biosorption of Ni was shown to be significantly higher at all other concentrations examined as compared to the lower concentration of Ni (10 mg L^−1^) and increased with increasing concentration. In comparison to other Ni concentrations, biosorption of 37.99, 33.74, and 30.84 mg L^−1^ were found for *B. subtilis*, *B. circulance*, and *B. coagulans* at 80 mg L^−1^, respectively. Herein, biosorption of Ni by the studied *Rhizobium* isolates followed the descending order of TAL–1148 > ICARDA 441 > ICARDA 39 > ICARDA 36 > ARC–207, and for *Bacillus* strains followed the descending order of *B. subtilis* > *B. circulance* > *B. coagulans* (Figure 1).

### 2.3. Pot Trial

#### 2.3.1. Parameters of the Growth

Depending on the concentration of Ni (0, 200, 400, and 600 mg kg^−1^) and bacterial inoculation (*Rhizobium* TAL–1148, *Rhizobium* TAL–1148 + *B. circulance*, *Rhizobium* TAL–1148 + *B. coagulans*, and *Rhizobium* TAL–1148 + *B. subtilis*), significant differences (*p* < 0.05) in the parameters of the growth of faba bean plants, i.e., dry weight, plant height, number of nodules, and N_2_%, were gathered during the course of two growing seasons (Table 2). During the 2019/2020 and 2020/2021 seasons, faba bean plants treated with T4 treatment (TAL–1148 + *B. subtilis*) showed significantly higher growth parameters than the plants that received the other treatments under 600 mg kg^−1^ Ni stress conditions, achieving 3.92 and 4.08 g plant^−1^, 40.10 and 41.01 cm plant^−1^, 69.33 and 74.33, and 2.72 and 2.79% for dry weight, plant height, number of nodules, and N_2_%, respectively (Table 2).

#### 2.3.2. Photosynthetic Pigments

At 60 days after sowing, the photosynthetic pigments (chlorophyll, carotenoids, and total soluble sugar) of faba bean leaves showed significant variations (*p* < 0.05) across different bacterial inoculation treatments: T1: *Rhizobium* TAL–1148 inoculation; T2: *Rhizobium* TAL–1148 + *B. circulance* inoculation; T3: *Rhizobium* TAL–1148 + *B. coagulans* inoculation; and T4: *Rhizobium* TAL–1148 + *B. subtilis* inoculation, at varied Ni stress concentrations (Figure 2).

Under 600 mg kg^−1^ Ni, the maximum value for total chlorophyll was 1.29, followed by 1.23 and 1.17 mg g^−1^ FW, and the highest value for total soluble sugar (TSS) was 3.69, followed by 3.56 and 3.40 µg g^−1^ FW for T4, followed by T3 and T2 treatments, over the control treatment (T1). However, T2 treatment was associated with the greatest value for carotenoids (0.37 µg g^−1^ FW) when compared to other treatments and the control in the 2019/2020 season (Figure 2). In the 2020/2021 season, a similar pattern was observed.

#### 2.3.3. Antioxidant Enzyme Activity

The activities of catalase (CAT), ascorbate peroxidase (APX), and polyphenol oxidase (PPO) were considerably altered as a result of bacterial inoculation treatments and Ni stress, as shown in Table 3. At 60 days after planting, varying amounts of Ni stress increased the amount of antioxidant enzyme activity in faba bean leaves compared to the control (Table 3).

Under different bacterial inoculation treatments, T4 treatment (inoculation with *Rhizobium* TAL–1148 + *B. subtilis*) efficiently increased the CAT content by 32.13 and 32.75 μM H_2_O_2_ g^−1^ FW min^−1^, APX content by 503.79 and 523.79 μM H_2_O_2_ g^−1^ FW min^−1^, and PPO content by 1.07 and 1.14 μM tetra-guaiacol g^−1^ FW min^−1^ during the first growing season (2019/2020), and the second growing season (2020/2021), respectively (Table 3). For diverse applications of bacterial inoculation treatments under Ni stress conditions, the results showed the following descending order: T4 > T2 > T3 > T1.

#### 2.3.4. Nickel Content

Table 4 shows that faba bean plants treated with bacterial inoculation had reduced Ni levels and accumulation in their tissues. In comparison to the 600 mg kg^−1^ Ni stress concentration, the T4 < T2 < T3 < T1 treatments attained 47.70, 50.90, 56.67, 97.71 µg g^−1^ for root contents and 24.28, 29.90, 38.67, 77.37 µg g^−1^ for shoot contents at 60 days after sowing in the 2019/2020 season, respectively. In the 2020/2021 season, a similar trend was observed.

These findings clearly suggest that treating faba bean plants with *Rhizobium* TAL–1148 + *B. subtilis* had a better effect than the other bacterial inoculation treatments due to the fact that the Ni content was lower. Under varied doses of Ni stress, the bioconcentration factor (BCF) and translocation factor (TF) of faba bean plants revealed that the application of *Rhizobium* TAL–1148 + *B. subtilis* (T4) considerably reduced the accumulation of Ni in plant tissues compared to the control treatment T1 (Table 4). 

## 3. Discussion

### 3.1. Assessment of Different Rhizobium Isolates and Bacillus Strains for Ni Tolerance

These differences in response shown by the studied strains might be due to differences in their inherent tolerance capacities, supported by active Ni efflux mechanisms to avoid dangerous intracellular Ni levels [19]. 

Several investigations have found that heavy metals, notably Ni, have a negative impact on symbiotic N fixation; for example, from the nodules of pea and lentil plants cultivated in polluted fields, Ni-tolerant *Rhizobium* strains (RP5 and RL9) were isolated and showed great tolerance to 350 and 500 mg mL^−1^ of Ni [20]. At the lowest dose of 0.2 mM, *Rhizobium* strains L9 and L19 showed better resistance to Ni than *Mesorhizobium* L42 and L50 [21]. In addition, in vitro, the rhizobium HGR-4 isolated from horse gram root nodules could tolerate 1000 mg g^−1^ Ni [14]. On the other hand, among the bacteria tested (*B. thuringiensis* 002, *B. fortis* 162, *B. subtilis* 174, and *B. farraginis* 354), *B. subtilis* 174 had the highest Ni tolerance, growing in conditions containing Ni at a concentration of 400 mg L^−1^ [13]. 

### 3.2. Biosorption of Ni by Different Rhizobium Isolates and Bacillus Strains

For the concentrations examined, the biosorption of Ni by different bacterial strains was significantly increased. The reason for this specific behavior is due to the smaller ionic radius of Ni (0.69 Å). In addition, bacteria can also accumulate metal in their cell walls, as well as protein polyphosphate complexes, polysaccharides, and complex forms with carboxyl groups of peptidoglycans [22]. Tobin et al. [23] hypothesized that molecules with a smaller ionic radius sorb more quickly. Biosorption of Ni has been well supported by previous findings based on ionic radius [24,25]. As a result, the aforementioned strains could be employed as potential heavy metal immobilizers in polluted soils. Ajmal et al. [26] reported that the bacterial strain *Citrobacter werkmanii* (WWN1) showed maximum net removal of 87% of Ni from an aqueous solution, followed by *Enterobacter cloacae* (JWM6), which showed 86% net removal of Ni, in a comparison with other studied strains.

### 3.3. Pot Trial

#### 3.3.1. Parameters of Growth

Rhizospheric bacteria have the ability to reduce/detoxify heavy metal stress through a variety of methods, such as metal ions outside the cell, biostimulation, bioaugmentation, metal reduction, and biosorption [27]. Improved plant development in metal-contaminated soils has been attributed to a bacterial biosorption/bioaccumulation mechanism with plant growth-promoting characteristics [28]. Metal accumulation in root nodules may be aided by rhizobial nodulation of the host plants. Additionally, different processes of precipitation, chelation, immobilization, and biosorption might lower metal toxicity when microbes remain in the rhizosphere [29]. Heavy metals such as Ni have a significant impact on plant nodulation growth parameters [30], and excessive Ni has been found to have negative effects on microorganisms, particularly rhizobia, and therefore on nodule formation in various leguminous species [31]. At 100 mg kg^−1^ Ni, more nodules were detected in *Vigna cylindrica*, while the production of root nodules was substantially hampered in *Vigna mungo* and *Vigna radiata* [32]. A phytotoxic effect was observed at 580 mg Ni/kg soil, which dramatically reduced the number of lentil nodules [20].

#### 3.3.2. Photosynthetic Pigments

Nickel almost completely destroys the photosynthetic apparatus/machinery, i.e., mesophyll cells and epidermal tissues [33], and reduces chlorophyll content (chlorophyll a, b, total chlorophyll) at all levels [34,35]. Furthermore, Ni affects the structure of thylakoid membranes and grana, lowering the size of grana and increasing the frequency of non-appressed lamellae [36]. However, higher levels of nutrients and organic matter in the rhizosphere could explain the rise in chlorophyll and carotenoids in faba bean leaves by bacterial inoculation [37,38]. Several studies have shown that bacterial inoculation accelerates the production of photosynthetic pigments in stressed plants [13,39,40,41].

#### 3.3.3. Activity of Antioxidant Enzymes

Plants enhance the activity of antioxidant enzymes in their main state in response to abiotic challenges, such as heavy metal stress; this is dependent on plant stress sensitivity as a first line of defense against high antioxidant ROS concentrations [42,43]. According to our findings, antioxidant enzymatic defense systems appear to play a key part in faba bean plant Ni toxicity. This defense can be activated at the transcriptional level, and at the enzymatic activity can help the plant adapt to Ni toxicity. A similar trend was observed in rye [44], *Lemna polyrhiza* [45], *Helianthus annus* [13], and lettuce [42]. 

#### 3.3.4. Nickel Content

Irrespective of Ni concentrations, the data showed that treating faba bean plants with *Rhizobium* TAL–1148 + *B. subtilis* had a better effect than other bacterial inoculation treatments due to the fact that the Ni content was lower. Under varied conditions of Ni stress, the bioconcentration factor (BCF) and translocation factor (TF) of faba bean plants revealed that the application of *Rhizobium* TAL–1148 + *B. subtilis* (T4) considerably reduced the accumulation of Ni in plant tissues compared to the control treatment, T1 (Table 4). 

Hence, the increase in Ni content in the roots of faba bean plants is due to biosorption of Ni by *Rhizobium* + *B. subtilis*. Based on these findings, it appears that biosorption by bacterial inoculation is responsible for the change of Ni into insoluble forms [14,42]. Reduced Ni levels in plant organs could be attributed to RL9 strain’s adsorption/desorption, according to research by the authors of [20,46]. The bioinoculant strains lowered Ni levels in the organs of inoculated plants cultivated in soils polluted with various metals [18,21].

## 4. Materials and Methods

### 4.1. Microorganisms and Growth Medium

The Department of Agricultural Microbiology, Soils, Water, and Environment Research Institute (SWERI), ARC, Egypt, provided five isolates of *Rhizobium leguminosarum* biovar *Viceae* (ICARDA 441, ICARDA 36, ICARDA 39, TAL–1148, and ARC–207), and three *Bacillus* strains (*B. subtilis* MF497446, *B. circulance* NCAIM B.02324, and *B. coagulans* NCAIM B.01123). These bacteria were grown in YEMB medium for *Rhizobium* isolates and NB medium for *Bacillus* strains, according to [47,48], respectively.

### 4.2. Assessment of Different Rhizobium Isolates and Bacillus Strains for Ni Tolerance

Nickel chloride (NiCl_2_.6H_2_O, Merck, Germany) was used to prepare a 1000 mg L^−1^ stock solution. In a shaker, 50 mL of YEMB and NB + 1 mL (10^8^ CFU mL^−1^) of fresh cultures of different strains and different levels of Ni (0, 10, 20, 40, 60, and 80 mg L^−1^) were built up and shaken at 150 rpm at 30 °C, then incubated for three days. Using a UV–Visible spectrophotometer (model 6705, Jenway, UK), the growth of bacteria was measured using optical density (OD) at 540 nm in five repetitions. A sterile uninoculated YEMB medium for *Rhizobium* isolates and NB media for *Bacillus* strains served as blanks. 

### 4.3. Biosorption of Ni by Different Rhizobium Isolates and Bacillus Strains

Experiments with initial concentrations of 0, 10, 20, 40, 60, and 80 mg L^−1^ were performed with 10^8^ CFU mL^−1^ of fresh cultures (30 °C and 150 rpm for 3 days) to evaluate the effect of varying Ni concentrations on biosorption by the different *Rhizobium* isolates and *Bacillus* strains. The bacterial cultures were then centrifuged for 10 min at 5000 rpm, with the supernatant filtered (5 mL) and examined with an Atomic Absorption Spectrophotometer (AAS PerkinElmer 3300). YEMB and NB broth, with the treated level of Ni and without inoculum, was used as a blank. The discrepancies between the first and last concentrations suggested that the studied bacteria absorbed Ni, and the experiment was repeated 3 times [49].

### 4.4. Pot Trial

Sandy soil was washed three times with 0.1 M HCl, then several times with distilled water to remove other minerals, sterilized twice for 4 h at 1.5 par and 121 °C, then mixed with different Ni concentrations and left for 2 weeks, after which 8 kg was placed into a polyethylene bag (22 cm in diameter and 35 cm in height) under greenhouse conditions during the 2019/2020 and 2020/2021 seasons [43]. With six repetitions, the experiment was performed according to a split-plot design. The main plots were the Ni pollution treatments (0, 200, 400, and 600 mg kg^−1^), while the inoculation treatments were subplots. There were four treatments in the subplots: (1) inoculation with *Rhizobium* (TAL–1148, control), (2) inoculation with *Rhizobium* (TAL–1148) + *B. circulance*, (3) inoculation with *Rhizobium* (TAL–1148) + *B. coagulans*, and (4) inoculation with *Rhizobium* (TAL–1148) + *B. subtilis*.

Surface sterilization of Faba bean seeds (*Vicia faba* C.V. Nobaria 1) was performed using alcohol 75% for 3 min, followed by 1 g L^−1^ HgCl_2_ solution for 2 min, and finally sterile water. In each pot, two seeds were sowed and irrigated twice-weekly using distilled water and fertilizer solution [50]. After germination, the pot was inoculated with 10 mL (1:1) from each culture (1 × 10^8^ CFU mL^−1^). 

#### 4.4.1. Trait Measurements

At 60 days following sowing, five healthy plants per treatment were uprooted, and growth parameters (dry weight (g plan^−1^), plant height (cm plant^−1^), number of nodules, and N_2_%) were measured. Plant dry weight was determined using an electronic scale, and the N_2_% was determined using the micro-Kejeldahl method, as previously described by [51]. Physiological properties (photosynthetic pigments, carotenoids, total soluble sugars), antioxidant enzyme activity (catalase (CAT), ascorbate peroxidase (APX), and poly phenol oxidase (PPO)), and Ni content in the roots and shoots of the faba bean plants were also studied. 

#### 4.4.2. Photosynthetic Pigments

To determine total chlorophyll and carotenoid contents, leaf samples (0.1 g) from each treatment were pulverized and extracted in 5 mL of acetone (80%), as described by [52]. The extract was detected at 663 nm, 645 nm, and 470 nm after centrifugation (13,000 *g* for 10 min). Carotenoid and total chlorophyll contents were calculated and expressed as mg g^−1^ FW. Following the protocol outlined in [53], total soluble sugars was determined. Leaf samples (0.5 g) from each treatment were homogenized in 5 mL ethanol (80%), then put in a water bath (80 °C for 30 min). After centrifugation (10,000 *g* for 10 min), the extract was collected and a UV spectrophotometer (Model 6705) was used to determine concentrations at 620 nm, based on a glucose standard curve and expressed as mg g^−1^ FW.

#### 4.4.3. Activity of Antioxidant Enzymes

Leaf samples (1 g) were homogenized in a chilled Tris–HCl buffer (0.1 mol L^−1^, pH 7.8) containing 1 mmol L^−1^ EDTA, 1 mmol L^−1^ dithiothreitol, and 5 mL polyvinyl pyrrolidone (4%) to estimate the activity of antioxidant enzymes. Using three replicates, ascorbate peroxidase (APX, μM H_2_O_2_ g^−1^ FW min^−1^), catalase (CAT, μM H_2_O_2_ g^−1^ FW min^−1^), and polyphenol oxidase (PPO, μM tetra-guaiacol g^−1^ FW min^−1^) were measured, according to [54,55,56], respectively. 

#### 4.4.4. Determination of Ni in the Roots and Shoots of the Faba Bean Plants

Plant roots and shoots were cleaned with distilled water, dried in an oven (70 °C for 24 h), and then ground in a stainless-steel blender, according to [57]. Then, 0.5 g of the ground samples was mixed with 4.0 mL HNO_3_ and 1.0 mL HClO_4_ and digested at 230 °C, then filtered to produce a clear solution. Flame atomic absorption spectroscopy was used to determine the overall concentration of Ni (AAS PerkinElmer 3300).

#### 4.4.5. Bioconcentration and Translocation Factors

For each plant component (roots and shoots), the Ni content efficiency of faba bean plants was measured. According to [58,59], the following equations were used to calculate the bioconcentration factor (BCF) and translocation factor (TF):$$\mathrm{BCF}\;=\frac{\mathrm{Concentration}\;\mathrm{of}\;\mathrm{Ni}\;\mathrm{in}\;\mathrm{roots}}{\mathrm{Concentration}\;\mathrm{of}\;\mathrm{Ni}\;\mathrm{in}\;\mathrm{test}\;\mathrm{soil}}\;\mathrm{TF}\;=\frac{\mathrm{Concentration}\;\mathrm{of}\;\mathrm{Ni}\;\mathrm{in}\;\mathrm{shoots}}{\mathrm{Concentration}\;\mathrm{of}\;\mathrm{Ni}\;\mathrm{in}\;\mathrm{roots}}$$

### 4.5. Statistical Analysis

Using CoStat software, the data were statistically evaluated using the analysis of variance (ANOVA) procedure (Pack-age 6.45, CoHort, USA). DMRT was used to compare the differences between the means at *p* < 0.01 and *p* < 0.05 [60]. The data are presented as means ± SDs.

## 5. Conclusions

The effects of several bacterial inoculations on Ni accumulation in faba bean plants grown in various levels of Ni-contaminated soil were studied. Inoculation with the most tolerant *Rhizobium* TAL–1148 + *B. subtilis* treatment was more effective in terms of growth parameters (dry weight of plant, plant height, number of nodules, and N_2_ content), as evidenced by physiological characteristics and antioxidant enzymes in soil treated with 600 mg kg^−1^ Ni compared to the other treatments. As a result, during the two growing seasons, the treatment combining the most tolerant *Rhizobium* (TAL–1148) with *B. subtilis* could be utilized as an option to reduce Ni toxicity.

## Figures and Tables

**Figure 1 plants-11-01812-f001:**
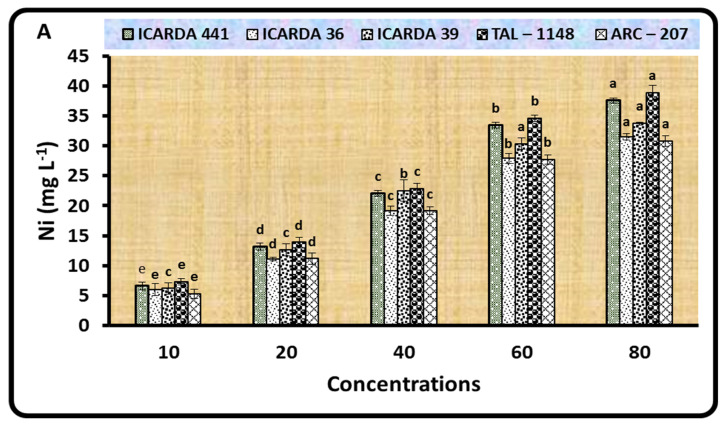
Biosorption of different concentrations of Ni (10, 20, 40, 60, and 80 mg L^−1^) by *Rhizobium* isolates (**A**) and *Bacillus* strains (**B**). According to Duncan’s test (*p* < 0.01), means followed by various letters show significant differences between the treatments. ^a–e^: Duncan’s letters.

**Figure 2 plants-11-01812-f002:**
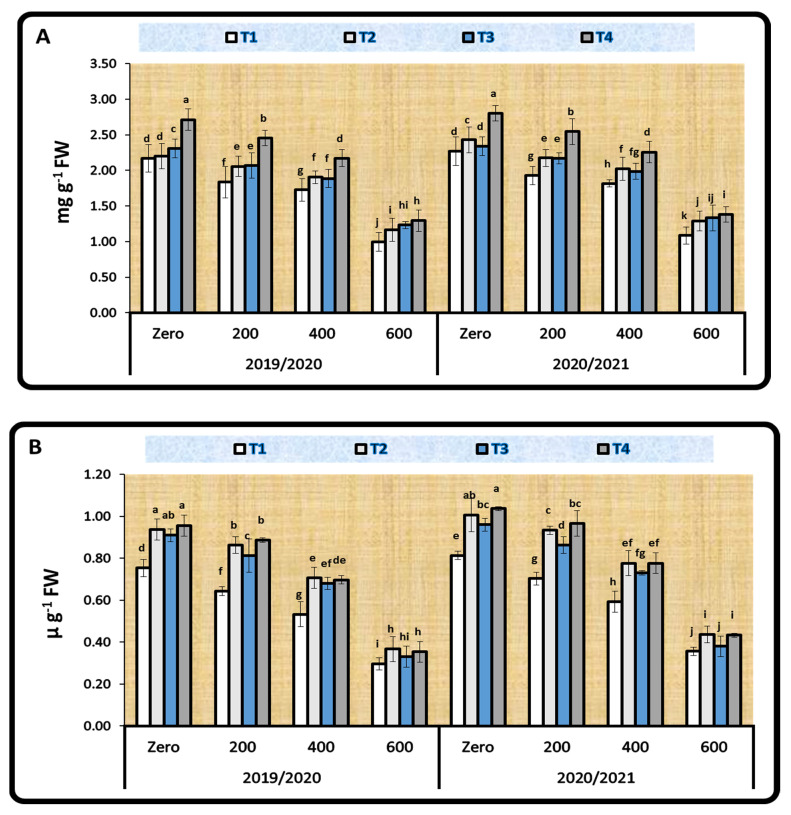
Effect of different concentrations of Ni and bacterial inoculation on total chlorophyll (**A**), carotenoids (**B**), and TSS (**C**) in faba bean leaves at 60 days after sowing during the 2019/2020 and 2020/2021 seasons. ^a–k:^ Duncan’s letters.

**Table 1 plants-11-01812-t001:** Growth patterns of *Rhizobium* isolates and *Bacillus* strains in different concentrations (0, 10, 20, 40, 60, and 80 mg L^−1^) of Ni.

Bacteria Test	Ni Concentrations (mg L^−1^)					
	0	10	20	40	60	80
** *Rhizobium* ** **isolates**						
**ICARDA 441**	4.83 ± 0.06 ^a^	4.30 ± 0.26 ^b^	3.70 ± 0.26 ^c^	3.00 ± 0.10 ^d^	2.60 ± 0.26 ^d^	1.84 ± 0.06 ^e^
**ICARDA 36**	4.87 ± 0.06 ^a^	4.20 ± 0.16 ^b^	3.67 ± 0.15 ^c^	3.17 ± 0.21 ^d^	2.94 ± 0.04 ^d^	1.57 ± 0.04 ^e^
**ICARDA 39**	4.87 ± 0.06 ^a^	4.43 ± 0.06 ^a^	3.40 ± 0.44 ^b^	2.40 ± 0.10 ^c^	1.60 ± 0.26 ^d^	0.82 ± 0.05 ^e^
**TAL–1148**	4.93 ± 0.15 ^a^	4.73 ± 0.15 ^a^	4.07 ± 0.15 ^b^	3.63 ± 0.15 ^c^	2.97 ± 0.15 ^d^	2.53 ± 0.06 ^e^
**ARC–207**	4.70 ± 0.02 ^a^	4.43 ± 0.59 ^a^	3.63 ± 0.15 ^b^	2.63 ± 0.10 ^c^	1.80 ± 0.10 ^d^	1.20 ± 0.10 ^d^
** *Bacillus* ** **strains**						
** *B. subtilis* **	2.83 ± 0.06 ^a^	2.70 ± 0.10 ^a^	2.33 ± 0.06 ^b^	1.67 ± 0.15 ^c^	1.13 ± 0.15 ^d^	0.84 ± 0.06 ^e^
** *B. circulance* **	2.87 ± 0.06 ^a^	2.20 ± 0.10 ^b^	1.67 ± 0.15 ^c^	1.17 ± 0.21 ^d^	0.94 ± 0.04 ^d^	0.57 ± 0.04 ^e^
** *B. coagulans* **	2.87 ± 0.06 ^a^	2.43 ± 0.06 ^b^	2.07 ± 0.15 ^c^	1.40 ± 0.10 ^d^	1.03 ± 0.12 ^e^	0.82 ± 0.05 ^e^

According to Duncan’s test (*p* < 0.01), means followed by various letters show significant differences between the treatments. The results are the averages and standard deviations (SDs) of three replicates. ^a–e^: Duncan’s letters.

**Table 2 plants-11-01812-t002:** Interaction effect of different concentrations of Ni and bacterial inoculation on the dry weight, plant height, number of nodules, and N% in faba bean plants at 60 days after sowing during the 2019/2020 and 2020/2021 seasons.

Treatments	Dry Weight (g Plant^−1^)	Plant Height (cm Plant^−1^)	Number of Nodules	N (%)
	**First Season (2019/2020)**			
**0 T1**	3.32 ± 0.55 ^h^	33.88 ± 1.48 ^h^	65.00 ± 5.00 ^f^	2.12 ± 0.15 ^h^
**0 T2**	4.17 ± 0.61 ^c^	42.59 ± 1.56 ^c^	74.00 ± 4.00 ^d^	2.97 ± 0.25 ^c^
**0 T3**	3.69 ± 0.64 ^e^	37.75 ± 1.41 ^e^	87.00 ± 6.00 ^b^	2.49 ± 0.14 ^e^
**0 T4**	4.46 ± 0.46 ^a^	45.56 ± 1.57 ^a^	98.00 ± 5.00 ^a^	3.26 ± 0.26 ^a^
**200 T1**	3.02 ± 0.60 ^j^	30.91 ± 1.41 ^j^	54.67 ± 4.51 ^h^	1.82 ± 0.14 ^j^
**200 T2**	3.98 ± 0.44 ^d^	40.63 ± 1.42 ^d^	63.67 ± 4.51 ^f^	2.78 ± 0.24 ^c^
**200 T3**	3.47 ± 0.53 ^fg^	35.45 ± 1.32 ^fg^	76.67 ± 4.51 ^c^	2.27 ± 0.23 ^fg^
**200 T4**	4.28 ± 0.70 ^b^	43.71 ± 1.46 ^b^	87.67 ± 4.51 ^b^	3.08 ± 0.14 ^b^
**400 T1**	2.87 ± 0.54 ^k^	29.31 ± 1.37 ^k^	43.67 ± 3.51 ^i^	1.67 ± 0.14 ^k^
**400 T2**	3.93 ± 0.62 ^d^	40.14 ± 1.19 ^d^	52.67 ± 3.51 ^h^	2.73 ± 0.22 ^c^
**400 T3**	3.45 ± 0.74 ^g^	35.28 ± 1.40 ^g^	65.67 ± 3.51 ^f^	2.25 ± 0.34 ^g^
**400 T4**	4.17 ± 0.54 ^c^	42.59 ± 1.37 ^c^	76.67 ± 3.51 ^c^	2.97 ± 0.24 ^c^
**600 T1**	2.53 ± 0.66 ^l^	25.85 ± 1.62 ^l^	25.33 ± 2.52 ^j^	1.33 ± 0.26 ^l^
**600 T2**	3.53 ± 0.75 ^f^	36.05 ± 1.56 ^f^	45.33 ± 3.06 ^i^	2.33 ± 0.15 ^f^
**600 T3**	3.16 ± 0.83 ^i^	32.28 ± 1.33 ^i^	58.33 ± 3.06 ^g^	1.96 ± 0.13 ^i^
**600 T4**	3.92 ± 0.44 ^d^	40.10 ± 1.40 ^d^	69.33 ± 3.06 ^e^	2.72 ± 0.24 ^c^
	**Second Season (2020/2021)**			
**0 T1**	3.53 ± 0.56 ^h^	34.76 ± 1.14 ^h^	68.00 ± 7.20 ^g^	2.25 ± 0.37 ^i^
**0 T2**	4.35 ± 0.49 ^c^	43.41 ± 1.90 ^c^	78.00 ± 2.10 ^e^	3.08 ± 0.27 ^c^
**0 T3**	3.91 ± 0.24 ^f^	38.60 ± 1.19 ^e^	90.00 ± 6.30 ^c^	2.58 ± 0.36 ^f^
**0 T4**	4.62 ± 0.66 ^a^	46.47 ± 1.35 ^a^	103.00 ± 4.20 ^a^	3.33 ± 0.52 ^a^
**200 T1**	3.23 ± 0.82 ^j^	31.79 ± 1.90 ^j^	57.67 ± 3.81 ^i^	1.95 ± 0.61 ^k^
**200 T2**	4.16 ± 0.94 ^d^	41.45 ± 1.95 ^d^	67.67 ± 3.71 ^g^	2.89 ± 0.49 ^d^
**200 T3**	3.69 ± 0.53 ^g^	36.30 ± 1.78 ^fg^	79.67 ± 6.71 ^de^	2.36 ± 0.48 ^h^
**200 T4**	4.44 ± 0.74 ^b^	44.62 ± 1.39 ^b^	92.67 ± 6.31 ^b^	3.15 ± 0.76 ^b^
**400 T1**	3.08 ± 0.94 ^k^	30.19 ± 1.62 ^k^	46.67 ± 7.11 ^k^	1.80 ± 0.92 ^l^
**400 T2**	4.11 ± 0.35 ^de^	40.96 ± 1.83 ^d^	56.67 ± 7.81 ^i^	2.84 ± 0.19 ^de^
**400 T3**	3.67 ± 0.63 ^g^	36.13 ± 1.78 ^g^	68.67 ± 3.61 ^g^	2.34 ± 0.37 ^h^
**400 T4**	4.33 ± 0.81 ^c^	43.50 ± 1.11 ^c^	81.67 ± 5.41 ^d^	3.04 ± 0.28 ^c^
**600 T1**	2.74 ± 0.93 ^l^	26.73 ± 1.77 ^l^	28.33 ± 4.32 ^l^	1.46 ± 0.17 ^m^
**600 T2**	3.71 ± 0.89 ^g^	36.87 ± 1.41 ^f^	49.33 ± 6.26 ^j^	2.44 ± 0.14 ^g^
**600 T3**	3.38 ± 0.49 ^i^	33.13 ± 1.48 ^i^	61.33 ± 6.16 ^h^	2.05 ± 0.51 ^j^
**600 T4**	4.08 ± 0.77 ^e^	41.01 ± 1.39 ^d^	74.33 ± 3.16 ^f^	2.79 ± 0.61 ^e^
**F-test**				
**Main**	******	******	******	******
**Sub main**	******	******	******	******
**Interaction**	******	******	******	******

According to Duncan’s test (*p* < 0.05), means followed by various letters show significant differences between the treatments. The results are the averages and standard deviations (SDs) of three replicates. Ni concentrations: **0**: 0 mg kg^−1^ of Ni; 200: 200 mg kg^−1^ of Ni; **400**: 400 mg kg^−1^ of Ni; **600**: 600 mg kg^−1^ of Ni. **T1**: inoculation with *Rhizobium (*TAL–1148); **T2**: inoculation with *Rhizobium (*TAL–1148) + *B. circulance*; **T3**: inoculation with *Rhizobium* (TAL–1148) + *B. coagulans*; **T4**: inoculation with *Rhizobium* (TAL–1148) + *B. subtilis*. **: Highly significant; ^a–m^: Duncan’s letters.

**Table 3 plants-11-01812-t003:** Interaction effect of different concentrations of Ni and bacterial inoculation on catalase (CAT, μM H_2_O_2_ g^−1^ FW min^−1^), ascorbate peroxidase (APX, μM H_2_O_2_ g^−1^ FW min^−1^) and polyphenol oxidase (PPO, μM tetra-guaiacol g^−1^ FW min^−1^) in faba bean leaves at 60 days after sowing during the 2019/2020 and 2020/2021 seasons.

Treatments	CAT	APX	PPO
	**First Season (2019/2020)**		
**0 T1**	10.41 ± 1.16 ^i^	275.21 ± 17.29 ^l^	0.35 ± 0.09 ^i^
**0 T2**	19.05 ± 1.59 ^ef^	393.07 ± 22.55 ^f^	0.63 ± 0.12 ^ef^
**0 T3**	16.64 ± 1.39 ^g^	347.36 ± 34.86 ^i^	0.55 ± 0.11 ^g^
**0 T4**	22.15 ± 1.98 ^d^	439.50 ± 41.67 ^d^	0.74 ± 0.13 ^d^
**200 T1**	14.13 ± 1.05 ^h^	315.93 ± 40.33 ^k^	0.47 ± 0.10 ^h^
**200 T2**	22.39 ± 1.91 ^d^	441.29 ± 32.23 ^d^	0.75 ± 0.12 ^d^
**200 T3**	18.20 ± 1.14 ^f^	382.71 ± 34.67 ^g^	0.61 ± 0.08 ^f^
**200 T4**	23.90 ± 2.65 ^c^	468.79 ± 41.33 ^c^	0.80 ± 0.05 ^c^
**400 T1**	14.47 ± 1.28 ^h^	334.86 ± 24.83 ^j^	0.48 ± 0.08 ^h^
**400 T2**	24.02 ± 2.64 ^c^	447.00 ± 43.91 ^d^	0.80 ± 0.09 ^c^
**400 T3**	18.67 ± 1.44 ^ef^	385.57 ± 38.46 ^fg^	0.62 ± 0.11 ^ef^
**400 T4**	27.00 ± 2.21 ^b^	482.00 ± 35.39 ^b^	0.90 ± 0.09 ^b^
**600 T1**	17.00 ± 1.10 ^g^	369.86 ± 25.67 ^h^	0.57 ± 0.02 ^g^
**600 T2**	27.54 ± 3.74 ^b^	470.93 ± 29.50 ^c^	0.92 ± 0.05 ^b^
**600 T3**	19.56 ± 2.47 ^e^	413.07 ± 34.83 ^e^	0.65 ± 0.11 ^e^
**600 T4**	32.13 ± 2.33 ^a^	503.79 ± 36.69 ^a^	1.07 ± 0.11 ^a^
	**Second season (2020/2021)**		
**0 T1**	11.07 ± 2.66 ^i^	289.21 ± 19.11 ^m^	0.41 ± 0.01 ^i^
**0 T2**	19.76 ± 1.09 ^ef^	414.07 ± 22.05 ^g^	0.68 ± 0.09 ^e^
**0 T3**	17.19 ± 1.19 ^g^	364.36 ± 39.26 ^j^	0.58 ± 0.08 ^g^
**0 T4**	22.77 ± 2.28 ^d^	459.50 ± 24.17 ^e^	0.81 ± 0.06 ^d^
**200 T1**	14.79 ± 2.31 ^h^	329.93 ± 34.63 ^l^	0.54 ± 0.09 ^h^
**200 T2**	23.10 ± 3.91 ^d^	462.29 ± 32.03 ^de^	0.80 ± 0.12 ^d^
**200 T3**	18.75 ± 1.67 ^f^	399.71 ± 29.57 ^h^	0.64 ± 0.10 ^f^
**200 T4**	24.52 ± 2.28 ^c^	488.79 ± 36.13 ^c^	0.87 ± 0.11 ^c^
**400 T1**	15.13 ± 1.90 ^h^	348.86 ± 27.55 ^k^	0.55 ± 0.08 ^gh^
**400 T2**	24.73 ± 2.74 ^c^	468.00 ± 44.08 ^d^	0.85 ± 0.09 ^c^
**400 T3**	19.22 ± 2.33 ^ef^	402.57 ± 42.12 ^h^	0.65 ± 0.01 ^ef^
**400 T4**	27.62 ± 3.88 ^b^	502.00 ± 45.25 ^b^	0.97 ± 0.08 ^b^
**600 T1**	17.66 ± 2.19 ^g^	383.86 ± 35.50 ^i^	0.63 ± 0.13 ^f^
**600 T2**	28.25 ± 2.94 ^b^	491.93 ± 25.29 ^c^	0.97 ± 0.08 ^b^
**600 T3**	20.11 ± 3.97 ^e^	430.07 ± 32.41 ^f^	0.68 ± 0.12 ^e^
**600 T4**	32.75 ± 3.09 ^a^	523.79 ± 46.31 ^a^	1.14 ± 0.14 ^a^
**F-test**			
**Main**	******	******	******
**Sub main**	******	******	******
**Interaction**	******	******	******

According to Duncan’s test (*p* < 0.05), means followed by various letters show significant differences between the treatments. The results are the averages and standard deviations (SDs) of three replicates. Ni concentrations: **0**: 0 mg kg^−1^ of Ni; 200: 200 mg kg^−1^ of Ni; **400**: 400 mg kg^−1^ of Ni; **600**: 600 mg kg^−1^ of Ni. **T1**: inoculation with *Rhizobium* (TAL–1148); **T2**: inoculation with *Rhizobium* (TAL–1148) + *B. circulance*; **T3**: inoculation with *Rhizobium* (TAL–1148) + *B. coagulans;* **T4**: inoculation with *Rhizobium* (TAL–1148) + *B. subtilis*. **: Highly significant; ^a–m^: Duncan’s letters.

**Table 4 plants-11-01812-t004:** Interaction effect of different concentrations of Ni and bacterial inoculation on the content of Ni in the roots and shoots, bioconcentration, and translocation factors in faba bean plants at 60 days after sowing during the 2019/2020 and 2020/2021 seasons.

Treatments	Ni Content in Root (µg g^−1^)	Ni Content in Shoots (µg g^−1^)	Bioconcentration Factor (BCF)	Translocation Factor (TF)
	**First Season (2019/2020)**			
**0 T1**	0.00	0.00	0.00	0.00
**0 T2**	0.00	0.00	0.00	0.00
**0 T3**	0.00	0.00	0.00	0.00
**0 T4**	0.00	0.00	0.00	0.00
**200 T1**	46.87 ± 6.16 ^e^	15.19 ± 1.87 ^fg^	0.23 ± 0.03 ^a^	0.32 ± 0.03 ^e^
**200 T2**	32.08 ± 2.75 ^g^	11.74 ± 2.47 ^g^	0.16 ± 0.09 ^d^	0.36 ± 0.07 ^e^
**200 T3**	33.54 ± 3.47 ^g^	15.27 ± 2.33 ^fg^	0.16 ± 0.05 ^cd^	0.45 ± 0.06 ^d^
**200 T4**	27.84 ± 4.33 ^h^	14.54 ± 1.98 ^fg^	0.13 ± 0.02 ^e^	0.52 ± 0.03 ^c^
**400 T1**	85.17 ± 7.66 ^b^	56.83 ± 4.30 ^b^	0.21 ± 0.05 ^b^	0.66 ± 0.04 ^b^
**400 T2**	42.03 ± 7.21 ^f^	17.69 ± 2.53 ^f^	0.10 ± 0.06 ^fg^	0.42 ± 0.06 ^e^
**400 T3**	47.49 ± 6.55 ^de^	21.82 ± 3.11 ^e^	0.11 ± 0.09 ^f^	0.45 ± 0.04 ^d^
**400 T4**	41.72 ± 4.55 ^f^	18.72 ± 3.67 ^f^	0.10 ± 0.01 ^fg^	0.44 ± 0.07 ^e^
**600 T1**	97.71 ± 5.50 ^a^	77.37 ± 3.95 ^a^	0.16 ± 0.05 ^c^	0.79 ± 0.01 ^a^
**600 T2**	50.90 ± 5.17 ^d^	29.90 ± 2.39 ^d^	0.08 ± 0.01 ^h^	0.58 ± 0.02 ^c^
**600 T3**	56.67 ± 4.17 ^c^	38.67 ± 2.27 ^c^	0.09 ± 0.01 ^gh^	0.68 ± 0.03 ^b^
**600 T4**	47.70 ± 6.57 ^de^	24.28 ± 2.69 ^e^	0.07 ± 0.01 ^h^	0.50 ± 0.02 ^d^
	**Second season (2020/2021)**			
**0 T1**	0.00	0.00	0.00	0.00
**0 T2**	0.00	0.00	0.00	0.00
**0 T3**	0.00	0.00	0.00	0.00
**0 T4**	0.00	0.00	0.00	0.00
**200 T1**	47.21 ± 4.33 ^d^	17.32 ± 1.20 ^f^	0.23 ± 0.01 ^a^	0.36 ± 0.01 ^e^
**200 T2**	34.47 ± 3.11 ^fg^	13.02 ± 2.41 ^g^	0.17 ± 0.03 ^d^	0.37 ± 0.03 ^e^
**200 T3**	34.32 ± 2.41 ^f^	15.61 ± 2.23 ^fg^	0.17 ± 0.02 ^d^	0.45 ± 0.04 ^d^
**200 T4**	30.40 ± 5.25 ^g^	15.99 ± 1.45 ^f^	0.15 ± 0.05 ^e^	0.52 ± 0.08 ^c^
**400 T1**	90.81 ± 4.17 ^b^	53.96 ± 4.13 ^b^	0.22 ± 0.02 ^b^	0.59 ± 0.09 ^b^
**400 T2**	44.42 ± 5.56 ^e^	16.97 ± 2.21 ^f^	0.11 ± 0.04 ^fg^	0.38 ± 0.05 ^e^
**400 T3**	50.27 ± 5.36 ^d^	23.16 ± 3.45 ^e^	0.12 ± 0.03 ^f^	0.46 ± 0.07 ^d^
**400 T4**	44.28 ± 5.78 ^e^	17.17 ± 3.72 ^f^	0.11 ± 0.02 ^fg^	0.38 ± 0.04 ^e^
**600 T1**	115.73 ± 4.32 ^a^	79.50 ± 3.54 ^a^	0.19 ± 0.02 ^c^	0.68 ± 0.05 ^a^
**600 T2**	53.46 ± 6.22 ^d^	27.35 ± 2.61 ^d^	0.08 ± 0.01 ^h^	0.51 ± 0.11 ^c^
**600 T3**	59.45 ± 5.09 ^c^	36.01 ± 2.52 ^c^	0.09 ± 0.02 ^gh^	0.60 ± 0.08 ^b^
**600 T4**	50.09 ± 4.47 ^d^	22.56 ± 2.86 ^e^	0.08 ± 0.01 ^h^	0.45 ± 0.06 ^d^
**F-test**				
**Main**	******	******	******	******
**Sub main**	******	******	******	******
**Interaction**	******	******	******	******

According to Duncan’s test (*p* < 0.05), means followed by various letters show significant differences between the treatments. The results are the averages and standard deviations (SDs) of three replicates. Ni concentrations: **0**: 0 mg kg^−1^ of Ni; 200: 200 mg kg^−1^ of Ni; **400**: 400 mg kg^−1^ of Ni; **600**: 600 mg kg^−1^ of Ni. **T1**: inoculation with *Rhizobium* (TAL–1148); **T2**: inoculation with *Rhizobium* (TAL–1148) + *B. circulance*; **T3**: inoculation with *Rhizobium* (TAL–1148) + *B. coagulans*; **T4**: inoculation with *Rhizobium* (TAL–1148) + *B. subtilis*. **: Highly significant; ^a–h^: Duncan’s letters.

## Data Availability

The study did not report any data.

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
