# Peer review of "Synergistic Interaction between Symbiotic N2 Fixing Bacteria and Bacillus strains to Improve Growth, Physiological Parameters, Antioxidant Enzymes and Ni Accumulation in Faba Bean Plants (Vicia faba) under Nickel Stress"

_plants, 2022, doi:10.3390/plants11141812_

Round 1

Reviewer 1 Report

The study analyzed Nickel stress tolerance and biosorption capability of rhizobiaand Bacillus strains in faba beans, knowing that soil contamination with heavy metals is a serious environmental issue, threatening agriculture sustainabilityfood safety, and public health

The experiment was well conceptualized and executed, presentation was clear. Following are some suggestions that can improve the paper:

- numbering of the chapters; as Discussion is #3, of course Methods should be 4, and Conclusion 5

- Introduction seems too long: the paragraph starting with “Heavy metal contamination...” should definitely be shortened and combined with previous paragraph starting with “Some stresses...”; same with the last two paragraphs of the Introduction

- move the explanations for Yeast Extract Mannitol Broth medium (YEMB) and Nutrient Broth (NB) from Methods to Results where they first appear

- if possible, use the same design for the two figures

- please explain the letters in each table and figure

Reviewer 2 Report

I recommend the publication of the article because scientific experimentation on the use of microorganisms in plants to increase nutrient uptake from the soil and disease resistance is of particular interest.

The aim and objectives of the article have been stated and are very interesting. The use of microorganisms to improve plant quality and production is an important topic, especially for the reduction of synthetic products in agriculture. The work done is certainly of international interest and the format applied is certainly suitable for a research article. The work is original, of particular interest and can certainly stimulate research on this topic. The length of the article is appropriate for the journal and the graphs and tables are clear and easy to understand. The conclusion summarises the aims of the work and future prospects.

Reviewer 3 Report

Important research results for science and agricultural practice. Two years of research, which I appreciate. I have included my comments in the text of the manuscript. After revising my comments, I recommend the manuscript for publication in Plants. The most important remarks are to correct: the Methodology and Methods chapter, Literature list and number the chapters correctly.
